# RELATIONAL SURROGATE LOSS LEARNING

**Tao Huang**[1][2]**, Zekang Li**[3]**, Hua Lu**[4]**, Yong Shan**[3]**, Shusheng Yang**[4]**,**
**Yang Feng**[3]**, Fei Wang**[5]**, Shan You**[2][*]**, Chang Xu**[1]
[1]School of Computer Science, Faculty of Engineering, The University of Sydney
[2]SenseTime Research
[3]Key Laboratory of Intelligent Information Processing,
  Institute of Computing Technology, Chinese Academy of Sciences (ICT/CAS)
[4]Huazhong University of Science and Technology
[5]University of Science and Technology of China

## ABSTRACT

Evaluation metrics in machine learning are often hardly taken as loss functions, as they could be non-differentiable and non-decomposable, *e.g.*, average precision and F1 score. This paper aims to address this problem by revisiting the surrogate loss learning, where a deep neural network is employed to approximate the evaluation metrics. Instead of pursuing an exact recovery of the evaluation metric through a deep neural network, we are reminded of the purpose of the existence of these evaluation metrics, which is to distinguish whether one model is better or worse than another. In this paper, we show that directly maintaining the relation of models between surrogate losses and metrics suffices, and propose a rank correlation-based optimization method to maximize this relation and learn surrogate losses. Compared to previous works, our method is much easier to optimize and enjoys significant efficiency and performance gains. Extensive experiments show that our method achieves improvements on various tasks including image classification and neural machine translation, and even outperforms state-of-the-art methods on human pose estimation and machine reading comprehension tasks. Code is available at: `https://github.com/hunto/ReLoss`.

## 1 INTRODUCTION

Evaluation metrics matter in machine learning since it depicts how well we want the models to perform. Nevertheless, most of them are non-differentiable and non-decomposable, thus we can not directly optimize them during training but resort to loss functions (or surrogate losses), which serve exactly as a proxy of task metrics. For example, pose estimation task uses percentage of correct keypoints (PCK) (Yang & Ramanan, 2012) to validate point-wise prediction accuracy, but it often adopts mean square error (MSE) as loss function. Neural machine translation task takes the sentence-level metric BLEU (Papineni et al., 2002) to evaluate the quality of predicted sentences, while using word-level cross-entropy loss (CE Loss) in training.

Besides this manual proxy, some works (Grabocka et al., 2019; Patel et al., 2020) propose to learn surrogate losses which approximate the metrics using deep neural networks (DNN), so the optimization of metrics can be relaxed to a differentiable space. For example, taking predictions and labels as input, (Grabocka et al., 2019) approximates the outputs of surrogate losses and evaluation metrics by minimizing their L2 distances. Moreover, recent work even involves the prediction networks into the surrogate loss learning by alternatively updating the loss and predictions, *i.e.*, they train the surrogate losses after every epoch during training, then use the latest optimized losses to train prediction networks in the next epoch. For instance, (Grabocka et al., 2019) mainly focuses on the simple binary classification while LS-ED (Patel et al., 2020) chooses to adopt the surrogate losses in the post-tuning stage (fine-tuning the models learned by original losses) and achieves promising improvements. However, these methods often suffer from heavy computational consumption and do not perform well on large-scale challenging datasets.

---

[*]Correspondence to: Shan You `<youshan@sensetime.com>`.

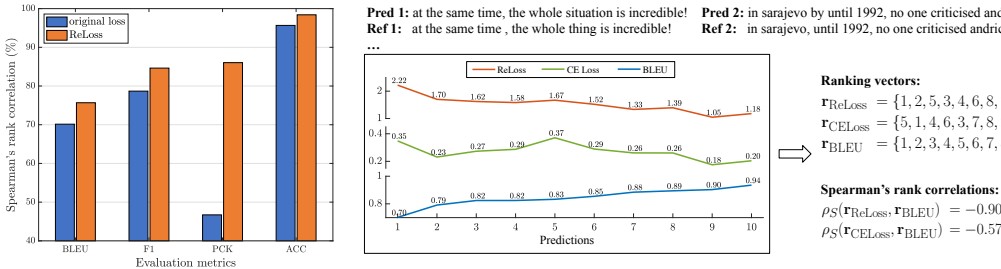

(a) Spearman's rank correlations     (b) Example on neural machine translation task

Figure 1: (a) Our proposed ReLoss significantly improves the ranking correlations between losses and metrics on various tasks. (b) Taking neural machine translation task as an example, we sample 10 sentences from WMT16 RO-EN dataset, then measure the BLEU, cross entropy (CE) loss, and ReLoss with trained network and ground-truth references. Compared to the original CE loss, our ReLoss obtains a stronger rank correlation.

Both manual and learned surrogate losses follow an *exact recovery* manner; namely, the surrogate losses should approximate the target metrics rigorously, and optimizing the surrogate loss is supposed to improve the evaluation metrics accordingly. However, this assumption does not always hold due to the approximation gap, bringing bias to the optimization and leading to sub-optimal results. Instead of pursuing an exact recovery of the evaluation metric, we are reminded of the purpose of metrics, which is to distinguish the performance of models. If a model has a smaller loss than the other model, its metric ought to be better. Nevertheless, current surrogate losses usually have weak relation with the evaluation metrics (*e.g.*, CE Loss & BLEU in Figure 1 (b)). Ideally, the surrogate loss should maintain strong relation of evaluation metric to all models.

In this paper, we leverage the ranking correlation as the relation between surrogate losses and evaluation metrics. Then a natural question raises, *if the loss functions only require accurate relative rankings to discriminate the models, why do we need to approximate the metrics exactly?* In this way, we propose a method named Relational Surrogate Loss (ReLoss) to maximize this rank correlation directly. Concretely, our ReLoss directly leverages the simple Spearman's rank correlation (Dodge, 2008) as the learning objective. By adopting differentiable ranking method, the ranking correlation coefficient can be maximized through gradient descent. Compared to exactly recovering the metrics, our correlation-based optimization is much easier to learn, and our ReLoss, which is simply constructed by multi-layer perceptions, aligns well with the metrics and obtains significantly better correlations compared to the original losses. For example, the commonly used loss MSE in pose estimation only has $46.71\%$ Spearman's rank correlation coefficient with the evaluation metric PCK, while our ReLoss enjoys $84.72\%$ relative improvement (see Table 1 and Figure 1).

Our ReLoss generalizes well to various tasks and datasets. We learn ReLoss using randomly generated data and pre-collected network outputs, then the learned losses are integrated into the training of prediction networks as normal loss functions (*e.g.*, cross-entropy loss), without any further fine-tuning. Note that we use the same surrogate losses with the same weights in each task, and we find that it is sufficient to obtain higher performance. Compared to previous works, our method is much easier to optimize and enjoys significant efficiency and performance improvements. Extensive experiments on the synthetic dataset and large-scale challenging datasets demonstrate our effectiveness. Moreover, our method outperforms the state-of-the-art methods in human pose estimation and machine reading comprehension tasks. For example, on human pose estimation task, our ReLoss outperforms the state-of-the-art method DARK (Zhang et al., 2020) by $0.2\%$ on COCO test-dev set; on machine reading comprehension task, we achieve new state-of-the-art performance on DuReader 2.0 test set, outperforming all the competitive methods, and even obtain $7.5\%$ better ROUGE-L compared to human performance.

## 2 RELATED WORK

**Surrogate loss learning.** Since most of the metrics in deep learning tasks are non-differentiable and non-decomposable (*e.g.*, accuracy, F1, AUC, AP, *etc.*), surrogate losses aim to approximate the metrics to make them differentiable using neural networks. (Grabocka et al., 2019) first proposes to learn surrogate losses by approximating the metrics of tasks through a neural network, and the losses

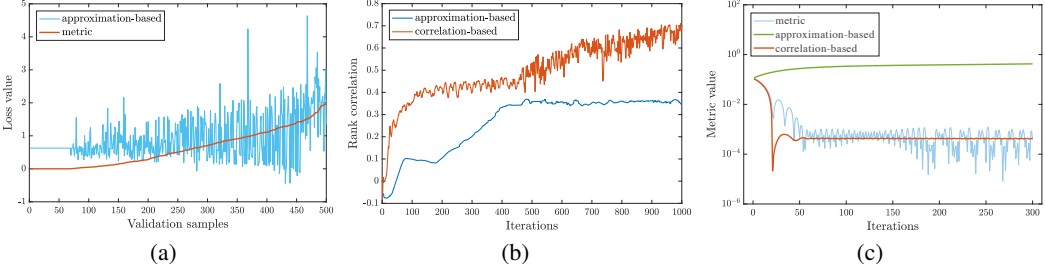

Figure 2: Visualization of our toy experiments on the synthetic dataset. (a) Visualization of outputs of approximation-based surrogate loss and evaluation metric on the validation set. (b) Curves of Spearmans' rank correlations between surrogate losses and evaluation metric in the training of losses. (c) Evaluation curves of different losses in training, lower is better.

are optimized jointly with the prediction model via bilevel optimization. (Patel et al., 2020) learns the surrogate losses via a deep embedding where the Euclidean distance between the prediction and ground truth corresponds to the value of the metric. However, it is hard to obtain a precise prediction by directly optimizing the surrogate loss with such a strong constraint. We remind that the role of loss functions is to determine which model is better, but with the unavoidable existence of approximation gap, this determinability does not always hold. In addition, these methods both train the surrogate losses alternately with prediction networks, resulting in noticeable efficiency and generability deduction compared to regular losses. In our paper, instead of only focusing on point-to-point recovery, which ignores the rankings between relative values of metrics, we ease the optimization constraint by explicitly learning our ReLoss with rank correlation, and enjoy significant performance and efficiency improvements.

**Differentiable sorting & ranking.** Differentiable sorting and ranking algorithms (Adams & Zemel, 2011; Grover et al., 2018; Blondel et al., 2020; Petersen et al., 2021) can be used in training neural networks with sorting and ranking supervision. Recent approach (Blondel et al., 2020) proposes to construct differentiable sorting and ranking operators as projections onto the permutahedron, *i.e.*, the convex hull of permutations, and using a reduction to isotonic optimization. (Petersen et al., 2021) proposes differentiable sorting networks by relaxing their pairwise conditional swap operations. In this paper, we can use any of these differentiable ranking algorithms to generate differentiable ranking vectors, then directly optimize the rank correlation coefficient for the supervision of our surrogate losses. The algorithm in (Petersen et al., 2021) is adopted for better performance.

## 3    PRELIMINARIES

For a given task with a metric function $\mathcal{M}(\boldsymbol{y}, \hat{\boldsymbol{y}})$, where $\boldsymbol{y}$ and $\hat{\boldsymbol{y}}$ denote the predicted labels and ground-truth labels, respectively, its loss function $\mathcal{L}(\boldsymbol{y}, \hat{\boldsymbol{y}})$ can be formulated as:

$$\mathcal{L}(\boldsymbol{y}, \hat{\boldsymbol{y}}) = f(\boldsymbol{y}, \hat{\boldsymbol{y}}), \tag{1}$$

where $f$ can be any function with output $\in \mathbb{R}^1$.

In this paper, we tend to use a learned DNN ($f_{\text{DNN}}$) with weights $\boldsymbol{\theta}_l$ as a surrogate loss, i.e.,

$$\mathcal{L}(\boldsymbol{y}, \hat{\boldsymbol{y}}; \boldsymbol{\theta}_l) = f_{\text{DNN}}(\boldsymbol{y}, \hat{\boldsymbol{y}}; \boldsymbol{\theta}_l). \tag{2}$$

The surrogate losses are learned with the networks' outputs $\boldsymbol{y}$ and the corresponding metric values $\mathcal{M}(\boldsymbol{y}, \hat{\boldsymbol{y}})$, i.e.,

$$\boldsymbol{\theta}_l^* = \arg\min_{\boldsymbol{\theta}_l} \mathcal{O}_\text{s}(\mathcal{L}(\boldsymbol{y}, \hat{\boldsymbol{y}}; \boldsymbol{\theta}_l), \mathcal{M}(\boldsymbol{y}, \hat{\boldsymbol{y}})), \tag{3}$$

where $\mathcal{O}_\text{s}$ is the learning objective of surrogate loss. The prediction networks with weights $\boldsymbol{\theta}_m$ are then optimized by descending the learned surrogate losses $\mathcal{L}(\boldsymbol{y}, \hat{\boldsymbol{y}}; \boldsymbol{\theta}_l^*)$, i.e.,

$$\boldsymbol{\theta}_m^* = \arg\min_{\boldsymbol{\theta}_m} \mathcal{L}(\boldsymbol{y}, \hat{\boldsymbol{y}}; \boldsymbol{\theta}_l^*). \tag{4}$$

**Approximation-based optimization.** To learn a surrogate loss w.r.t. a metric, an intuitive idea is to approximate the metric's outputs, *i.e.*, learn the surrogate losses by minimizing the distances

between the outputs of surrogate losses and their corresponding metric values, which is adopted in previous works (Grabocka et al., 2019; Patel et al., 2020), their learning objective $\mathcal{O}_s$ is

$$\mathcal{O}_s(\mathcal{L}(\boldsymbol{y}, \hat{\boldsymbol{y}}; \boldsymbol{\theta}_l), \mathcal{M}(\boldsymbol{y}, \hat{\boldsymbol{y}})) = \parallel \mathcal{L}(\boldsymbol{y}, \hat{\boldsymbol{y}}; \boldsymbol{\theta}_l) - \mathcal{M}(\boldsymbol{y}, \hat{\boldsymbol{y}}) \parallel_2^2, \qquad (5)$$

we call this optimization as approximation-based optimization.

However, it is hard for a DNN to fully recover the evaluation metric. We conduct toy experiments using a random weighted DNN with output $\in \mathbb{R}^1$ as an evaluation metric, and then the surrogate loss is learned using limited observations of the metric. As illustrated in Figure 2 (a), since approximating a random network with random inputs is challenging, the errors between surrogate loss learned by approximation-based optimization and metric values are noticeably large. In order to validate the effectiveness of losses in training, we then train the input data with metric or learned losses, as shown in Figure 2 (c), we illustrate the curves of metric values w.r.t. learned input data during training, and directly using metric as loss function obtains best metric value (lower is better), but the performance of input data using approximation-based loss is getting worse.

# 4 LEARNING RELATIONAL SURROGATE LOSS

## 4.1 RELATION AS RANK CORRELATION

Based on the previous discussion, the prior works adopt an unnecessary constraint by enforcing the surrogate losses to fully recover the evaluation metrics. However, the loss function only needs to have the same ranking relation to the metrics, *i.e.*, we just need to make the surrogate losses have the same ranking as metrics. In this paper, we obtain the relation between surrogate losses and evaluation metrics by using rank correlation as the learning objective, which we call correlation-based optimization.

The relation between surrogate losses and evaluation metrics is measured by ranking correlation, which is a statistic that measures the relationship between rankings of the same variable. A ranking correlation coefficient measures the degree of similarity between two rankings and can be used to assess the relation's significance. If the surrogate loss fully correlates to the evaluation metric, the descent of loss value will always obtain better metric values.

**Spearman's rank correlation.** For optimization of surrogate losses, we use the most commonly used Spearman's rank correlation (Dodge, 2008). For two vectors $\boldsymbol{a}$ and $\boldsymbol{b}$ with size $n$, the Spearman's rank correlation is defined as:

$$\rho_S(\boldsymbol{a}, \boldsymbol{b}) = \frac{\mathrm{Cov}(\mathbf{r}_{\boldsymbol{a}}, \mathbf{r}_{\boldsymbol{b}})}{\mathrm{Std}(\mathbf{r}_{\boldsymbol{a}})\mathrm{Std}(\mathbf{r}_{\boldsymbol{b}})} = \frac{\frac{1}{n-1}\sum_{i=1}^{n}(\mathbf{r}_{\boldsymbol{a}i} - E(\mathbf{r}_{\boldsymbol{a}}))(\mathbf{r}_{\boldsymbol{b}i} - E(\mathbf{r}_{\boldsymbol{b}}))}{\mathrm{Std}(\mathbf{r}_{\boldsymbol{a}})\mathrm{Std}(\mathbf{r}_{\boldsymbol{b}})}, \qquad (6)$$

where $\mathbf{r}_{\boldsymbol{a}}$ is the rank vector of $\boldsymbol{a}$, $\mathrm{Cov}(\mathbf{r}_{\boldsymbol{a}}, \mathbf{r}_{\boldsymbol{b}})$ is the covariance of the rank vectors, $\mathrm{Std}(\mathbf{r}_{\boldsymbol{a}})$ denotes the standard derivation of $\mathbf{r}_{\boldsymbol{a}}$.

## 4.2 LEARNING LOSSES BY MAXIMIZING RANK CORRELATION

**Correlation-based optimization.** We use Spearman's rank correlation as the objective to learn our surrogate losses, since the loss should have a negative correlation w.r.t. the metric (higher is better), our objective is to minimize the Spearman's rank correlation coefficient, *i.e.*,

$$\mathcal{O}_s(\mathcal{L}(\boldsymbol{y}, \hat{\boldsymbol{y}}; \boldsymbol{\theta}_l), \mathcal{M}(\boldsymbol{y}, \hat{\boldsymbol{y}})) = \rho_S(\mathcal{L}(\boldsymbol{y}, \hat{\boldsymbol{y}}; \boldsymbol{\theta}_l), \mathcal{M}(\boldsymbol{y}, \hat{\boldsymbol{y}})) \qquad (7)$$

Since the computation of rank vectors $\mathbf{r}_{\boldsymbol{a}}$ and $\mathbf{r}_{\boldsymbol{b}}$ in Eq.(6) is not differentiable, we adopt one of the differentiable ranking methods (Petersen et al., 2021) to obtain differentiable ranking vectors, and empirically find that the errors in differentiable approximation is negligible and our learned correlation can be very close to the optimal value, *i.e.*, $\mathcal{O}_s = -1$.

As shown in Figure 2, compared to approximation-based optimization, the surrogate loss learned by our correlation-based optimization obtains higher rank correlation and faster convergent speed. Besides, optimizing with our correlation-based loss achieves significantly better performance than approximation-based optimization, and is more stable than the original loss (evaluation metric).

**Learning with Gradient Penalty.** In our paper, we can directly backward through the surrogate loss to obtain its gradients. However, we find the first-order derivative of the learned ReLoss w.r.t. the prediction $y$ changes rapidly since we only constrain the correlation in Eq.(3), which result in either vanishing or exploding gradients. Nevertheless, in the optimization of networks, we want a loss with smooth gradients to train the networks steadily.

Following (Gulrajani et al., 2017; Patel et al., 2020), we now propose an alternative way to smooth the gradients by enforcing the Lipschitz constraint. A differentiable function is 1-Lipschitz if and only if it has gradients with norm at most 1 everywhere, so we consider directly constraining the gradient norm of the loss's output w.r.t. its input, *i.e.*

$$\mathcal{L}_{\text{penalty}} = (\| \nabla_y \mathcal{L}(y, \hat{y}; \theta_l) \|_2 - 1)^2. \tag{8}$$

This penalty of gradients has been shown to enhance the training stability for generative adversarial networks (Gulrajani et al., 2017). Our objective of surrogate loss learning in Eq.(7) becomes

$$\mathcal{O}_s(\mathcal{L}(y, \hat{y}; \theta_l), \mathcal{M}(y, \hat{y})) = \rho_S(\mathcal{L}(y, \hat{y}; \theta_l), \mathcal{M}(y, \hat{y})) + \lambda \mathcal{L}_{\text{penalty}}, \tag{9}$$

we use $\lambda = 10$ in our experiments.

### 4.3 PIPELINE

Now we illustrate how to learn our ReLoss. Different from previous works (Grabocka et al., 2019; Patel et al., 2020) which train the surrogate loss and prediction network alternatively as bilevel optimization, we want our surrogate loss to be general as vanilla loss (*e.g.*, cross-entropy loss). Since we learn the surrogate loss with a much weaker constraint, our surrogate loss can generalize better to the whole distribution of outputs and metric values to train the surrogate loss once for all, without further fine-tuning.

Our training strategy of surrogate loss is summarized in Algorithm 1. The training data of surrogate losses is the combination of randomly generated data $G_R$ and the outputs of models $G_M$. Concretely, we design a random generator to produce random outputs and labels uniformly for $G_R$, while for $G_M$, we use the intermediate checkpoints of the prediction networks trained by original loss to predict the outputs of train data. Each batch of training data is generated from $G_R$ or $G_M$ with probabilities $p$ and $1 - p$, respectively.

---

**Algorithm 1** Learning of surrogate losses.

**Input:** surrogate loss $\mathcal{L}$ with random weights $\theta_l$, batch size $N$, metric function $M$, data generators $G_M$ and $G_R$, sample probability $p$.
**Output:** learned surrogate loss with highest correlation.
1: **while** *not converged* **do**
2:     $L = \emptyset$ ; $M = \emptyset$ ; $L_p = \emptyset$ ;
3:     **for** $i = 1, .., N$ **do**
4:         generate a batch of predictions and ground-truth labels $(y_i, \hat{y}_i)$ from $G_R$ with probability $p$ or $G_M$ with probability $1 - p$ ;
5:         compute loss *w.r.t.* predictions and labels: $l_i = \mathcal{L}(y_i, \hat{y}_i; \theta_l)$ ;
6:         compute metric: $m_i = \mathcal{M}(y_i, \hat{y}_i)$ ;
7:         compute $l_{pi} = (\| \nabla_y \mathcal{L}(y_i, \hat{y}_i; \theta_l) \|_2 - 1)^2$ ;
8:         $L = L \cup \{l_i\}$ ; $M = M \cup \{m_i\}$ ; $L_p = L_p \cup \{l_{pi}\}$ ;
9:     **end for**
10:    $\mathcal{L}_{\text{penalty}} = \frac{1}{N} \sum_{i=1}^{N} L_p$ ;
11:    optimize $\theta_l$ by descending $\nabla_{\theta_l}(\rho_S(L, M) + \lambda \mathcal{L}_{\text{penalty}})$ ;
12: **end while**
13: **return** learned surrogate loss with weights $\theta_l^*$ .

---

**Usage of learned ReLoss.** The learned ReLoss can be fixed and then integrated into the training of prediction networks, *i.e.*, we only change the loss function in training, without any modification on training strategy, network architecture, *etc.*Besides, we emprically find that ReLoss would achieve better performance if combined with the regular loss. In this case, the regular loss might act as a regularization term, and bring a decent prior for the ReLoss to enhance the optimization.

Table 1: Rank correlations between loss function (descending order) and metrics (ascending order) in different tasks, higher is better.

| Task | Metric | Original loss | Spearman's (%) | | Kendall's Tau (%) | |
|---|---|---|---|---|---|---|
| | | | origin | ReLoss (ours) | origin | ReLoss (ours) |
| Classification | ACC | CE | 95.66 | 98.40 (+2.74) | 83.69 | 89.88 (+6.19) |
| Human Pose Estimation | PCK | MSE | 46.71 | 86.04 (+39.33) | 33.04 | 69.00 (+35.96) |
| Machine Reading Comprehension | F1 | CE | 78.68 | 84.63 (+5.95) | 61.49 | 67.96 (+6.47) |
| Neural Machine Translation | BLEU | CE | 70.14 | 75.68 (+5.54) | 65.37 | 70.17 (+4.80) |

Table 2: Results on CIFAR-10, CIFAR-100, and ImageNet datasets.

| Dataset | Model | CE | | ReLoss | |
|---|---|---|---|---|---|
| | | Top-1 (%) | Top-5 (%) | Top-1 (%) | Top-5 (%) |
| CIFAR-10 | ResNet-56 | $94.32 \pm 0.25$ | - | $\mathbf{94.57} \pm 0.08$ | - |
| CIFAR-100 | ResNet-56 | $73.61 \pm 0.11$ | - | $\mathbf{74.15} \pm 0.14$ | - |
| ImageNet | ResNet-50 | 76.5 | 93.0 | **76.8** | 93.0 |
| | MobileNet V2 | 71.8 | 90.3 | **72.2** | 90.5 |

## 5 EXPERIMENTS

To fully experiment with the effectiveness and generability of our ReLoss, we conduct experiments on both computer vision and natural language processing tasks. In computer vision, we experiment on image classification and human pose estimation tasks, while in natural language processing, we experiment on machine reading comprehension and neural machine translation tasks.

We first show the rank correlations of original losses and our learned surrogate losses to the metrics in Table 1. Spearman's (Dodge, 2008) and Kendall's Tau (Kendall, 1938) are two commonly used coefficients to measure the ranking correlations between two vectors. We can see that in all our experimented tasks, our ReLoss achieves higher correlations compared to the original losses. Notably, even for cross-entropy (CE) loss, which has been shown to align well with the misclassification rate, our surrogate loss still performs better on classification tasks with the metric accuracy (ACC).

### 5.1 COMPUTER VISION

**Image classification.** We conduct experiments on three benchmark datasets CIFAR-10, CIFAR-100 (Krizhevsky et al., 2009), and ImageNet (Deng et al., 2009). On CIFAR-10 and CIFAR-100 datasets, we train ResNet-56 (He et al., 2016) with original CE loss and our surrogate loss and report their accuracies on the test set with mean and standard derivation of 5 runs. While on ImageNet dataset, we train ResNet-50 and MobileNet V2 (Sandler et al., 2018), their accuracies on validation set are reported. Notably, all experiments use the same surrogate loss with the same weights.

Table 2 shows the evaluation results. We can see that, though the original CE loss obtains a very high correlation ($\sim 0.96$ in Table 1), by integrating our surrogate loss with higher correlation, the performance can still be improved. Note that we use the same surrogate loss with fixed weights in these three datasets, which means that our loss can generalize to different image classification datasets and gain the improvements with negligible additional cost.

**Human pose estimation.** Human pose estimation (HPE) aims to locate the human body and build body skeleton from images. It is difficult to precisely evaluate the performance of HPE since many features need to be considered (*e.g.*, the quality of body parts, the precision of each keypoints). As a result, many metrics are proposed for HPE. Percentage of correct keypoints (PCK) (Yang & Ramanan, 2012) and Average Precision (AP) are two of the most commonly used ones. However, current methods usually adopt mean square error (MSE) to minimize the distance between predicted heatmap and target heatmap, which correlates weakly with the evaluation metrics.

In our experiments, we choose to approximate PCK@0.05 since it better reflects the quality of each keypoint, and our ReLoss achieves significant improvement on rank correlation compared to the original MSE loss. We use the most widely used large-scale dataset COCO (Lin et al., 2014) to evaluate our performance, and the results are summarized in Table 3. We can see that, on validation set, our ReLoss significantly improves the baseline methods, and the $AP^{75}$ improves the most since

Table 3: Results of human pose estimation task on COCO dataset.

| Method | Backbone | Input size | AP | $AP^{50}$ | $AP^{75}$ | $AP^M$ | $AP^L$ | AR | PCK@0.05 |
|---|---|---|---|---|---|---|---|---|---|
| validation set | | | | | | | | | |
| SimpleBaseline (Xiao et al., 2018) | ResNet-50 | $256 \times 192$ | 70.4 | 88.6 | 78.3 | 67.1 | 77.2 | 76.3 | 85.0 |
| SimpleBaseline + ReLoss | ResNet-50 | $256 \times 192$ | **71.9** | **89.9** | **80.0** | **68.0** | **77.9** | **77.3** | **86.1** |
| HRNet (Sun et al., 2019) | HRNet-W32 | $256 \times 192$ | 74.4 | **90.5** | 81.9 | 70.8 | 81.0 | 79.8 | 86.7 |
| HRNet + ReLoss | HRNet-W32 | $256 \times 192$ | **74.8** | **90.5** | **82.4** | **70.9** | **81.2** | **79.9** | **87.3** |
| test-dev set | | | | | | | | | |
| G-RMI (Papandreou et al., 2017) | ResNet-101 | $353 \times 257$ | 64.9 | 85.5 | 71.3 | 62.3 | 70.0 | 69.7 | - |
| SimpleBaseline (Xiao et al., 2018) | ResNet-101 | $384 \times 288$ | 73.7 | 91.9 | 81.1 | 70.3 | 80.0 | 79.0 | - |
| HRNet (Sun et al., 2019) | HRNet-W48 | $384 \times 288$ | 75.5 | 92.5 | 83.3 | 71.9 | 81.5 | 80.5 | - |
| DARK (Zhang et al., 2020) | HRNet-W48 | $384 \times 288$ | 76.2 | 92.5 | 83.6 | 72.5 | 82.4 | 81.1 | - |
| DARK + ReLoss | HRNet-W48 | $384 \times 288$ | **76.4** | **92.7** | **83.7** | **72.7** | **82.5** | **81.3** | - |

our ReLoss aligns PCK for better keypoint localization. On test-dev set, we integrate our ReLoss into state-of-the-art method DARK (Zhang et al., 2020) and achieve improvements on all the metrics.

## 5.2 NATURAL LANGUAGE PROCESSING

The gaps between loss functions and evaluation metrics on natural language processing tasks are severer since the tasks often use sentence-level evaluation metrics (*e.g.*, BLEU and ROUGE-L) but adopt word-level cross-entropy loss in training.

**Machine reading comprehension.** The task of machine reading comprehension (MRC) aims to empower machines to answer questions after reading articles. Concretely, with a given question, the models are required to locate a segment of text from the corresponding reading passage, which is most probably the answer. We use F1 score as the evaluation metric to learn surrogate loss, and experiment on two typical MRC datasets SQuAD (Rajpurkar et al., 2016) and DuReader (He et al., 2018). SQuAD evaluates performance using F1-score, and DuReader uses ROUGE-L (Lin, 2004) and BLEU-4 (Papineni et al., 2002).

The evaluation results are summarized in Table 4 and Table 5. On DuReader 2.0 dataset, our ReLoss gains improvements on dev set, and achieves state-of-the-art performance on test set. On SQuAD 1.1 dataset, we also achieve improvements compared to the baseline method.

Table 4: Results of machine reading comprehension task on DuReader 2.0 dataset. †: reported by (He et al., 2018).

| Method | ROUGE-L | BLEU-4 | F1 |
|---|---|---|---|
| dev set | | | |
| MacBERT-base (Cui et al., 2020) | 51.4 | 50.3 | 53.9 |
| MacBERT-base + ReLoss | **51.8** | **50.6** | **54.2** |
| MacBERT-large (Cui et al., 2020) | 53.2 | 51.2 | 55.5 |
| MacBERT-large + ReLoss | **53.6** | **51.4** | **55.9** |
| test set | | | |
| BiDAF† (Seo et al., 2016) | 39.2 | 31.9 | - |
| Wang et al. (2018) | 44.2 | 41.0 | - |
| MCR-Net-large (Peng et al., 2021) | 50.8 | 49.2 | - |
| Human Performance† | 57.4 | 56.1 | - |
| MacBERT-large + ReLoss | **64.9** | **61.8** | - |

Table 5: Results on SQuAD 1.1 dataset compared with BERT (Devlin et al., 2018).

| Method | F1 | EM |
|---|---|---|
| BERT-base | 88.5 | 80.8 |
| BERT-base + ReLoss | **88.8** | **81.3** |
| BERT-large | 90.9 | 84.1 |
| BERT-large + ReLoss | **91.4** | **84.6** |

**Neural machine translation.** Neural machine translation (NMT) aims to translate a sentence from the source to the target language with an end-to-end neural model. The evaluation metric of NMT is BLEU (Papineni et al., 2002), which measures the n-gram overlap between the generated translation and the reference. We conduct experiments on the Non-Autoregressive neural machine Translation (NAT) task, in which the model generates target words independently and simultaneously. Since the

Table 6: Evaluation results of BLEU on Neural Machine Translation task. We report the performance of our methods on the WMT16 EN-RO dataset. Transformer denotes the auto-regressive model. * denotes the performance that we reproduced using the public code.

| Model | Speed | Original loss | | ReLoss on EN-RO | | ReLoss on RO-EN | |
|---|---|---|---|---|---|---|---|
| | | EN-RO | RO-EN | EN-RO | RO-EN | EN-RO | RO-EN |
| Transformer (Vaswani et al., 2017) | $1.0\times$ | 32.88 | 33.94 | - | - | - | - |
| NAT-Base (Gu et al., 2017) | $15.6\times$ | 29.24 | 28.97 | 30.07 +0.83 | 29.68 +0.71 | 29.93 +0.69 | 29.61 +0.64 |
| BoN-$L_1$(N=2)* (Shao et al., 2021) | $15.6\times$ | 30.76 | 30.46 | 30.96 +0.20 | 30.74 +0.28 | 30.88 +0.12 | 30.78 +0.32 |

output of NAT cannot be properly evaluated through word-level cross-entropy loss due to the multi-modality problem in language, the correlation between cross-entropy loss and translation quality is weak, limiting the NAT performance.

Table 6 shows the NAT evaluation results on WMT-16 EN→RO and RO→EN datasets. We conduct experiments based on the NAT-base and a strong baseline BoN-$L_1$ (N=2) (Shao et al., 2021), which introduce the BoN loss to fine tune NAT-base by modeling the bag of ngrams in the sentence. We integrate ReLoss into the baseline methods NAT-base (Gulrajani et al., 2017) and BoN-$L_1$ (N=2), and the evaluation results show that our ReLoss can improve both of them.

## 5.3 ABLATION STUDIES

**Compare with LS-ED.** Prior work LS-ED (Patel et al., 2020) aims to post-tune the scene text recognition (STR) model using a surrogate loss, which is learned with approximation-based optimization. In order to compare our method with LS-ED, we conduct experiments on the same settings. Following LS-ED, we learn the surrogate loss using edit distance, then fine-tune the trained model using our learned loss (without using original loss for fair comparisons). The results in Table 7 show that our ReLoss significantly outperforms the baselines CE and LS-ED. Note that we only train our ReLoss once then integrate it into training, indicating that our loss is more efficient and general.

Table 7: Evaluation results on scene text recognition task comparing with CE and LS-ED. The reported metrics are accuracy (ACC, higher is better), normalized edit distance (NED, higher is better), and total edit distance (TED, lower is better).

| Test dataset | ↑ACC (%) | | | ↑NED | | | ↓TED | | |
|---|---|---|---|---|---|---|---|---|---|
| | CE | LS-ED | ReLoss | CE | LS-ED | ReLoss | CE | LS-ED | ReLoss |
| IIIT-5K (Mishra et al., 2012) | 87.500 | **87.933** | 87.700 | 0.961 | **0.963** | 0.961 | 722 | **645** | 667 |
| SVT (Wang et al., 2011) | 87.172 | 86.708 | **87.481** | 0.952 | 0.954 | **0.957** | 180 | 163 | **156** |
| ICDAR'03 (Lucas et al., 2005) | 94.302 | 94.535 | **94.579** | 0.979 | 0.981 | **0.982** | 110 | 99 | **98** |
| ICDAR'13 (Karatzas et al., 2013) | 92.020 | 92.299 | **92.709** | 0.966 | 0.979 | **0.981** | 137 | 108 | **101** |
| ICDAR'15 (Karatzas et al., 2015) | **78.520** | 78.410 | 78.355 | 0.915 | 0.915 | **0.916** | 868 | **837** | 845 |
| SVTP (Phan et al., 2013) | 78.605 | 79.225 | **80.310** | 0.912 | 0.913 | **0.915** | 346 | 333 | **316** |
| CUTE (Risnumawan et al., 2014) | 73.171 | 74.216 | **75.958** | 0.871 | 0.875 | **0.884** | 224 | 219 | **195** |
| Wins | 1 | 1 | **5** | 0 | 1 | **6** | 0 | 2 | **5** |
| Average | 84.470 | 84.761 | **85.299** | 0.937 | 0.940 | **0.943** | 370 | 343 | **340** |

**Transferability of learned ReLoss.** In all our experiments, we use the same surrogate loss in each task. If we learn different surrogate losses on specific datasets, would the performance be better? To validate this, we conduct experiments to train ReLoss independently on each dataset, as shown in Table 9. The ReLoss transferred from ImageNet dataset performs similar to the consistent ReLoss learned on corresponding datasets. It might be because we train the ReLoss using predicted and randomly generated data, and it is sufficient to cover different distributions of datasets on image classification. We also experiment on NMT task. As shown in Table 6, we train ReLoss on both EN→RO and RO→EN, and the results using either of them to train the networks are similar, which demonstrates that the learned ReLoss is language-independent and can bring similar improvements on the other translation direction.

**Comparison of approximation-based and our correlation-based optimization.** In Figure 2, we compare our ReLoss with approximation-based methods on the synthetic dataset. Now we further conduct experiments on image classification task to show our superiority. Concretely, we learn the

surrogate losses with the same architecture using approximation-based or our correlation-based optimization, then integrate them to train networks on CIFAR datasets. As shown in Table 8, our ReLoss with correlation-based optimization obtains the highest accuracies compared to the CE loss and approximation-based loss. Note that the standard derivations of accuracies of approximation-based loss are much larger than CE loss and correlation-based loss; this might be because the imprecise rankings and gradients in approximation-based loss weaken the training stability.

Table 8: Results of different optimization methods on image classification task.

| Loss function | Rank correlation (%) | | ACC (%) | |
|---|---|---|---|---|
| | Spearman's | Kendall's Tau | CIFAR-10 | CIFAR-100 |
| Cross Entropy | 95.66 | 83.69 | $94.32 \pm 0.25$ | $73.61 \pm 0.11$ |
| ReLoss (approximation-based) | 91.71 | 76.03 | $94.11 \pm 0.42$ | $73.88 \pm 0.32$ |
| ReLoss (correlation-based) | **98.40** | **89.88** | $\mathbf{94.57} \pm 0.08$ | $\mathbf{74.15} \pm 0.14$ |

**Integrating ReLoss with / without regular losses.** We empirically find that the prediction networks using our ReLoss converge very fast at the beginning of training, then the performance will increase very slowly or even get worse, the experiments on synthetic dataset show the similar trend (see Figure 2 (c)). A possible reason is that there exist some data points that surrogate losses can not predict accurately, making the optimization fall into local minima. For better performance, we use the regular loss as a regularization term to help the surrogate losses jump out local minima. We conduct experiments to show the differences by integrating ReLoss with or without regular loss. As summarized in Table 10, the performance drops if not adding regular loss in training, showing that the regular losses can bring a decent prior for ReLoss to achieve better performance.

Table 9: Comparison of transferred ReLoss and consistent ReLoss.

| Dataset | ACC (%) | |
|---|---|---|
| | transferred | consistent |
| CIFAR-10 | $94.57 \pm 0.08$ | $94.61 \pm 0.12$ |
| CIFAR-100 | $74.15 \pm 0.14$ | $74.12 \pm 0.09$ |

Table 10: Evaluation results w/ or w/o regular losses.

| Dataset | ACC (%) | |
|---|---|---|
| | w/ regular loss | w/o regular loss |
| CIFAR-10 | $94.57 \pm 0.08$ | $93.82 \pm 0.32$ |
| CIFAR-100 | $74.15 \pm 0.14$ | $73.91 \pm 0.22$ |
| ImageNet | 76.8 | 75.9 |

## 5.4 COMPLEXITY ANALYSIS

Denoting the training iterations of surrogate losses as $T_l$, the training epochs and iterations in each epoch of prediction networks are $E_m$ and $T_m$, respectively, the runtime complexity of our ReLoss is $\mathcal{O}(T_l + E_m \times T_m)$. For comparison, the runtime complexity of regular loss is $\mathcal{O}(E_m \times T_m)$, while for previous surrogate loss learning method (Grabocka et al., 2019), it trains the surrogate losses after every iteration and has a runtime complexity of $\mathcal{O}(E_m \times T_l + E_m \times T_m)$. Note that our additional cost $\mathcal{O}(T_l)$ of learning ReLoss costs only 0.5 GPU hour on image classification with a single NVIDIA TITAN Xp GPU, and we only need to train ReLoss once for each task, reducing much computational cost compared to previous works.

## 6 CONCLUSION

As a proxy of the evaluation metric, loss function matters in machine learning since it controls the optimization of networks. However, it is often hard to design a loss function with strong relation to the evaluation metric. In this paper, we aim to address this problem by learning surrogate losses using deep neural networks. Unlike previous works that pursue an exact recovery of the evaluation metric, we are reminded of the essence of the loss function and evaluation metric, which is to distinguish the performance of models, and show that directly maximizing the rank correlation between surrogate loss and evaluation metric can learn better loss. How to design and learn a more robust and general surrogate loss would be a valuable aspect to improve this work.

ACKNOLEDGEMENTS

Chang Xu was supported by the Australian Research Council under Project DP210101859 and the University of Sydney SOAR Prize.

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

# A APPENDIX

## A.1 NEURAL ARCHITECTURES OF OUR SURROGATE LOSSES

**Image classification.** In order to make the learned surrogate loss generalize to all the classification tasks which take accuracy as the metric, we use the logits $y_{\text{pos}}$ with positive labels as the input of our neural network, which is the same as cross-entropy loss, and the outputs of the surrogate loss are simply computed through 4-layer perceptions with intermediate activations, *i.e.*,

$$l = \text{Mean}(\text{FC}(\text{ELU}(\text{FC}(\text{ELU}(\text{FC}(\text{ELU}(\text{FC}(y_{\text{pos}})))))))), \tag{10}$$

we use ELU activation (Clevert et al., 2015) for stable gradients since it is $\text{C}^{\infty}$ continuous.

**Machine reading comprehension and neural machine translation.** Since the evaluation metrics of MRC and NMT are computed by a sequence of texts, based on the architecture in image classification, we use additional self-attention mechanisms (Vaswani et al., 2017) to extract sequential information.

**Human pose estimation.** Given the prediction heatmap and target heatmap, the original MSE loss is used to minimize the distance between these two heatmaps. Our ReLoss first embeds these two heatmaps into two hidden vectors, then computes the MSE loss between them as the final loss.

## A.2 TRAINING STRATEGIES

**Surrogate loss.** We train the surrogate losses using Adam optimizer with a fixed learning rate of 0.01, and the weight decay is set to 1e-4.

**Image classification.** The reported models are trained using the same code, with the only difference in the loss function. On CIFAR-10 and CIFAR-100 datasets, we train ResNet-20 for 200 epochs with an initial learning rate of 0.1, which decays 0.1 at 100th and 150th epochs, the batch size is set to 128 with cutout (DeVries & Taylor, 2017) data augmentation, we run each experiment 5 times with different random seeds and report their mean accuracy with standard derivation. On ImageNet, we follow the same training strategy as in torchvision[1] (Marcel & Rodriguez, 2010). Concretely, we train ResNet-50 for 120 epochs with an initial learning rate of 0.1, a step learning rate scheduler which decays 0.1 every 30 epochs is adopted. While for MobileNet V2, we train it for 300 epochs with 4e-5 weight decay, a cosine learning rate scheduler is adopted with an initial learning rate of 0.045. The batch sizes for ResNet-50 and MobileNet V2 are both set to 32. We use SGD optimizer with 0.9 momentum on all datasets. Note that all the experiments use the same surrogate loss with the same weights.

**Human pose estimation.** We train ResNet-50, HRNet-W32, and DARK-HRNet-W48 following the default configurations in MMPose (Contributors, 2020). Concretely, the models are trained with Adam optimizer for 210 epochs, and a step learning rate scheduler is adopted with initial value 5e-4, which decays 0.1 at 170th and 200th epochs. The total batch sizes of 8 GPUs with input size $256 \times 192$ and $384 \times 288$ are 512 and 256, respectively.

**Machine reading comprehension.** We train the networks using Adam optimizer with weight decay 0.01, a linear learning rate strategy which warmups 0.1 epoch and decays 2 epochs is adopted. On DuReader 2.0 dataset, the batch size is set to 32; we train MacBERT-base and MacBERT-large with learning rates 3e-5 and 2e-5, respectively. On SQuAD 1.1 dataset, we train BERT-base and BERT-large with batch sizes 32 and 2, and the learning rates are 5e-5 and 1e-5, respectively.

**Neural machine translation.** For WMT16 EN-RO, we use the WMT 2016 corpus, which consists of 610K sentence pairs for training. We take news-dev-2016 and news-test-2016 as development and test sets. We learn a joint BPE model with 32K operations and share the vocabulary for source and target languages. As knowledge distillation (Hinton et al., 2015; Kim & Rush, 2016) has been proven to be crucial for training NAT models, we first train an auto-regressive transformer model (Vaswani et al., 2017) as the teacher and then apply sequence-level knowledge distillation to construct the corpus for training NAT models. The NAT-base takes the same architecture as the base transformer model except that we modify the attention mask of the decoder for not masking the future tokens. We use a target length predictor to predict the length of the target sentence. We use golden length

---

[1]https://github.com/pytorch/vision/tree/main/references/classification

during the training and the predicted length during the inference. For training ReLoss, we only use the outputs of the model. For NAT-base, the number of training steps is 200K. We select the checkpoint based on the validation set. We add ReLoss with a factor of 1 on the CE loss to fine-tune the NAT-base for 10k steps with a batch size of 32 and a fixed learning rate of 1e-5. For BoN-L1 (N=2) (Shao et al., 2021), we reproduce the results using the public repo [2]. We combine ReLoss with the BoN Loss to fine-turn the model for 3K steps with a batch size of 512, which keeps the same with the BoN-L1 (N=2).

## A.3 MORE ABLATION STUDIES

**Performance of losses with different rank correlations.** To show the influence of the rank correlations on the performance, we choose the surrogate losses with different rank correlations to train models on CIFAR-10 dataset. The results in Figure 3 (a) clearly show that the increase of rank correlation boosts the performance, and the losses with lower rank correlations will disturb the training of networks.

**Performance of losses with different network capacities.** Our surrogate losses are constructed with fully connected layers. To validate the influence of capacities of loss model, we conduct experiments to learn losses on different capacities on CIFAR-10 dataset. As shown in Figure 3 (b), our original loss model has 33.4K parameters, we adjust the number of layers or hidden dimensions to change the network capacity. The results show that the performance of our ReLoss gets saturated on a small number of parameters. With this small network capacity, its computational cost is negligible in the training of prediction networks.

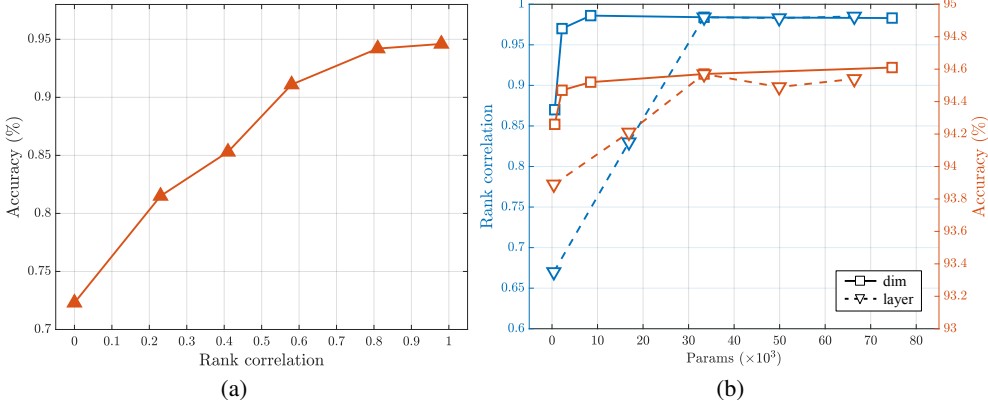

Figure 3: (a) Evaluation results on CIFAR-10 using ReLoss with different rank correlations. (b) Evaluation results on CIFAR-10 using different capacities of surrogate losses.

Table 11: Compare with alternate training on image classification.

| Loss function | ACC (%) | | |
|---|---|---|---|
| | CIFAR-10 | CIFAR-100 | ImageNet |
| Cross Entropy | $94.32 \pm 0.25$ | $73.61 \pm 0.11$ | 76.4 |
| ReLoss | $94.57 \pm 0.08$ | $74.15 \pm 0.14$ | 76.8 |
| ReLoss (alternate training) | $\mathbf{94.65} \pm 0.21$ | $\mathbf{74.18} \pm 0.31$ | **76.9** |

**Compare with alternately learning losses.** The outputs of prediction networks change during the training. For surrogate losses using approximation-based optimization, it is hard to obtain accurate predictions for all the possible predictions, so the previous works (Grabocka et al., 2019; Patel et al., 2020) learn surrogate losses alternately with prediction networks during training. In our paper, we show that our ReLoss achieves higher performance using only pre-trained surrogate losses. We further conduct experiments to train our ReLoss alternately with prediction networks. As shown in Table 11, the alternate training obtains higher accuracies but gets more unstable on CIFAR datasets. We think this might be because the update of weights in surrogate losses will disturb the gradients,

---

[2]https://github.com/ictnlp/Seq-NAT

while our ReLoss without alternate training provides the same gradients for the same predictions and labels, thus more stable. Since the performance improvements of alternate training are marginal, we can use pre-trained losses for better generalization and efficiency.

**Effect of gradient penalty.** Our paper aims to stabilize the gradients of surrogate losses by introducing a gradient penalty regularization in loss learning. We conduct experiments to show the effectiveness of the gradient penalty. As summarized in Table 12, we train the models on CIFAR datasets using the surrogate losses with or without gradient penalty. The results show that the ReLoss without gradient penalty performs poorly compared to the one with gradient penalty and even the original loss, although it obtains a good rank correlation. It indicates that the regularization of gradients of the surrogate losses is necessary and contributes a lot to the performance.

Table 12: Results of ReLoss with or without gradient penalty.

| Loss function | Rank correlation (%) | | ACC (%) | |
|---|---|---|---|---|
| | Spearman's | Kendall's Tau | CIFAR-10 | CIFAR-100 |
| Cross Entropy | 95.66 | 83.69 | $94.32 \pm 0.25$ | $73.61 \pm 0.11$ |
| ReLoss (w/o gradient penalty) | 98.31 | 89.56 | $94.28 \pm 0.31$ | $73.03 \pm 0.26$ |
| ReLoss (w/ gradient penalty) | **98.40** | **89.88** | $\mathbf{94.57 \pm 0.08}$ | $\mathbf{74.15 \pm 0.14}$ |

As for the weight $\lambda$ in Eq.(9), we empirically find that this regularization is easy to achieve since our learning objective of correlation is weak. We have tried different values of $\lambda$, the term of gradient penalty is always very small ($\sim$1e-3), so we directly follow previous work (Gulrajani et al., 2017) and use $\lambda = 10$.

**Robustness of performance in multiple independent runs.** To validate the robustness of the training of ReLoss, we conduct experiments to train the ReLoss multiple times independently, and leverage these learned surrogate losses to train ResNet-56 on CIFAR datasets. As the results summarized in Table 13, the accuracies of multiple runs are similar (with low standard variance), showing that our ReLoss can obtain stable results. We believe that our ReLoss is easy to learn, and the regularization term of gradient penalty could obtain stable gradients of surrogate losses w.r.t. the logits. As a result, the performance would be robust.

Table 13: Results of ReLoss on CIFAR datasets in multiple independent runs.

| Number | Rank correlation (%) | | ACC (%) | |
|---|---|---|---|---|
| | Spearman's | Kendall's Tau | CIFAR-10 | CIFAR-100 |
| 1 | 98.36 | 89.91 | $94.49 \pm 0.06$ | $74.05 \pm 0.09$ |
| 2 | 98.47 | 89.78 | $94.51 \pm 0.08$ | $74.12 \pm 0.11$ |
| 3 | 98.43 | 89.86 | $94.43 \pm 0.10$ | $74.09 \pm 0.08$ |
| 4 | 98.42 | 89.75 | $94.59 \pm 0.08$ | $74.15 \pm 0.10$ |
| 5 | 98.38 | 89.71 | $94.55 \pm 0.07$ | $74.11 \pm 0.06$ |
| mean $\pm$ std | $98.41 \pm 0.04$ | $89.80 \pm 0.08$ | $94.51 \pm 0.06$ | $74.10 \pm 0.03$ |

**Compare with rank-based classification loss.** Our method adopts differentiable sort algorithms (Blondel et al., 2020; Petersen et al., 2021) to train the surrogate loss. However, Blondel et al., 2020 proposes a rank-based classification loss to directly calculate the L1 error between predicted soft ranks and target ranks of top-1 elements, i.e.,

$$l_{\text{rk}} = |\boldsymbol{r}_{\text{pos}} - N|, \tag{11}$$

where $\boldsymbol{r}_{\text{pos}}$ denotes the predicted soft ranks of $\boldsymbol{y}_{\text{pos}}$ and $N$ is the number of classes.

We train the above rank-based classification loss (RankLoss) on CIFAR-10, CIFAR-100, and ImageNet datasets with the same models and strategies in our paper. From the results summarized in Table 14, we can see that the accuracies obtained by RankLoss are significantly lower than the original CE loss and our ReLoss. Besides, on datasets with more classes (100 and 1000 on CIFAR-100 and ImageNet, respectively), it only obtains slightly better accuracies than random guess. One possible reason is that RankLoss only focuses on the ranks of positive elements of logits, lacking supervision on the remained elements. As a result, the network receives little information to converge on datasets with large numbers of classes. In contrast, our ReLoss learns from the evaluation metrics and supplies better information for discriminating models, thus achieves better accuracy.

On the other hand, RankLoss is hard to generalize to different tasks since it needs to design different loss functions for different metrics. Meanwhile, it cannot be applied to regression tasks. As a result, we believe it is necessary to learn a metric-oriented surrogate loss using a neural network rather than directly applying differentiable ranking operators as the loss.

Table 14: Compare with rank-based classification loss.

| Dataset | Model | CE Loss (%) | RankLoss (%) | ReLoss (%) |
|---|---|---|---|---|
| CIFAR-10 | ResNet-56 | 94.32 | 82.77 | **94.57** |
| CIFAR-100 | ResNet-56 | 73.61 | 5.65 | **74.15** |
| ImageNet | ResNet-50 | 76.5 | 0.58 | **76.8** |

**GPU memory and training cost compared to original loss.** We report the memory consumption and training speed of ResNet-56 on CIFAR-10 and CIFAR-100 datasets in Table 15. Since our ReLoss only has $\sim 0.03$M parameters, the memory and training time increments are negligible compared to the much larger consumptions of models (e.g., ResNet-50 has 25.6M parameters).

Table 15: Comparisons of GPU memory and training cost.

| Dataset | Loss function | GPU memory (M) | Training speed (batches / second) | ACC (%) |
|---|---|---|---|---|
| CIFAR-10 | CE | 618.46 | 14.29 | 94.32 |
| | ReLoss | 618.68 | 14.29 | **94.57** |
| CIFAR-100 | CE | 618.55 | 14.27 | 73.61 |
| | ReLoss | 618.76 | 14.27 | **74.15** |

## A.4 VISUALIZATION OF CONVERGENCE CURVES IN TRAINING

We visualize the convergence curves of CE loss and our surrogate loss in Figure 4. We can see that our loss obtains higher validation accuracies over the whole training procedure.

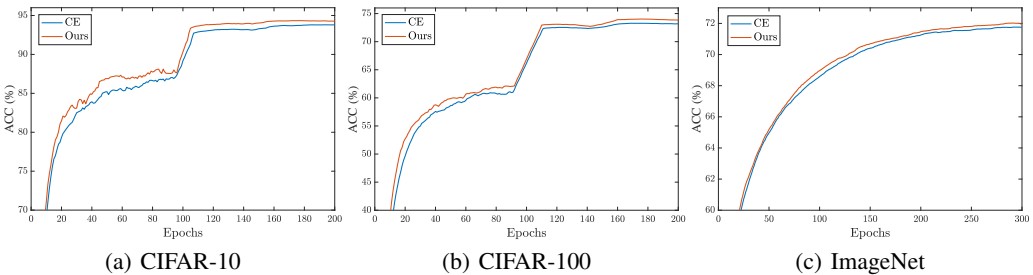

(a) CIFAR-10          (b) CIFAR-100          (c) ImageNet

Figure 4: Convergence curves (validation accuracies) of CE loss and our loss on CIFAR-10, CIFAR-100, and ImageNet datasets. The data is smoothed using a moving average with a factor $0.25$. Zoom up to view better.

