# OpenReview forum: "Relational Surrogate Loss Learning"
_ICLR.cc/2022/Conference — ICLR 2022 Poster_

### Official Review · Reviewer_L49G · 2021-11-02

**Correctness:** 4
**Technical Novelty And Significance:** 4
**Empirical Novelty And Significance:** 4
**Recommendation:** 8
**Confidence:** 4

**Main Review:**

Pros:
* The idea is novel and interesting. Loss functions matter a lot to the performance, while the current hand-craft losses often align poorly to the performance, surrogate loss learning is a way to alleviate this issue. Instead of directly approximating the evaluation metrics as previous methods, this paper proposes a new learning method by revisiting the purpose of loss functions, which is to distinguish the performance of models. Hence, the authors aim to learn the surrogate losses by making the surrogate losses have the same discriminability as the evaluation metrics. The idea is straightforward and is easy to implement by using ranking correlation as an optimization objective.
* To obtain stable gradients, the authors involve a gradient penalty regularization term by enforcing 1-Lipschitz, which is effective according to the experiments in Appendix.
* The experimental results look good. By keeping the same backbones and training strategies, the proposed ReLoss gains improvements on various tasks, including CV and NLP tasks, and outperforms the state-of-the-art methods on human pose estimation and machine reading comprehension tasks.
* The comparisons are sufficient to show the proposed method's superiority. The authors conduct experiments on the synthetic and large-scale benchmark datasets, and gain significant improvements in performance and efficiency compared to existing surrogate loss learning methods.

Cons:
* The ReLoss is learned by optimization, which means the qualities of losses may be different in multiple runs. How many times did the authors train the ReLoss to obtain the performance reported in the paper? It would be better to conduct experiments to show the performance of surrogate losses learned in multiple independent runs.


**Summary Of The Paper:**

This paper proposes a surrogate loss learning method named ReLoss. The ReLoss learned by maximizing the relation between surrogate losses and evaluation metrics is used to replace the original losses. Extensive experiments on computer vision (CV) tasks (image classification, pose estimation, and scene text recognition) and natural language processing (NLP) tasks (machine reading comprehension and translation) are provided, showing the benefits.

**Summary Of The Review:**

In summary, the reviewer thinks the paper gives an effective way to obtain better losses, the experiments are well-conducted, and even improve the state-of-the-art methods, which would be helpful to the community. I would recommend it for acceptance.

---

> ### Author Response · Authors · 2021-11-22
> **Response to Reviewer L49G**
>
> Thanks for your efforts in reviewing our paper. The responses to your comments are as follows.
>
> **Q1: It would be better to conduct experiments to show the performance of surrogate losses learned in multiple independent runs.**
>
> Thanks for your valuable suggestion. We have updated the paper and added the results of our performance on CIFAR-10 using the ReLoss learned in multiple independent runs (see Table 13 in Appendix). A summary of the results is as below:
>
> |Number|Spearman's correlation (%)|Kendall's Tau correlation (%)|CIFAR-10 ACC (%)|CIFAR-100 ACC (%)|
> |:--:|:--:|:--:|:--:|:--:|
> |1|98.36|89.91|94.49 $\pm$ 0.06|74.05 $\pm$ 0.09|
> |2|98.47|89.78|94.51 $\pm$ 0.08|74.12 $\pm$ 0.11|
> |3|98.43|89.86|94.43 $\pm$ 0.10|74.09 $\pm$ 0.08|
> |4|98.42|89.75|94.59 $\pm$ 0.08|74.15 $\pm$ 0.10|
> |5|98.38|89.71|94.55 $\pm$ 0.07|74.11 $\pm$ 0.06|
> |mean $\pm$ std|98.41 $\pm$ 0.04|89.80 $\pm$ 0.08|94.51 $\pm$ 0.06|74.10 $\pm$ 0.03|
>
> According to the results, the accuracies of multiple runs are similar with low standard variance, showing that our ReLoss can obtain stable results. We believe that our ReLoss is easy to learn, and the regularization term of gradient penalty could obtain stable gradients of surrogate losses w.r.t the logits. As a result, the performance would be robust.

---

> > ### Comment · Reviewer_L49G · 2021-12-02
> > **Thanks for your response**
> >
> > Thanks for your response. The listed minor issue has been clarified, so I will keep my rating

---

### Official Review · Reviewer_eBZN · 2021-11-02

**Correctness:** 4
**Technical Novelty And Significance:** 4
**Empirical Novelty And Significance:** 4
**Recommendation:** 6
**Confidence:** 4

**Main Review:**

- The study field of this paper is important. According to the authors’ experiments in Table 1, the original manually-designed losses do not align well with the evaluation metrics on some tasks. The proposed ReLoss can improve those ranking correlations a lot.

- The proposed method can be widely applied to NLP and CV tasks, and its benefits seem significant according to the experiments. E.g., on human pose estimation task, ReLoss outperforms the state-of-the-art method with only replacement of loss function.

Minor Weakness:

- How much additional costs if we train a model using ReLoss? How much time does it cost to train ReLoss? Would ReLoss increase GPU memory or training time in training? This is related to the practical feasibility of the proposed loss.



**Summary Of The Paper:**

- This paper proposes a relational surrogate loss learning method (ReLoss) inspired by the fact that the evaluation metric and loss are used to distinguish whether one model is better or worse than another.

- This paper provides extensive experiments that demonstrate the effectiveness of the proposed method. The performance and efficiency compared to existing surrogate loss methods are significant, and the performance compared to original losses is seem to be significant on various tasks.


**Summary Of The Review:**

I think the proposed method is well-principled and provides meaningful improvements on various tasks. I would like to lean on the positive side for this paper.

---

> ### Author Response · Authors · 2021-11-22
> **Response to Reviewer eBZN**
>
> Thanks for taking the time to review our paper! The responses to your comments and questions are as follows.
>
> **Q1: The additional time cost to train the ReLoss.**
>
> **A1**: Thanks for your helpful question. Compared to training with the original loss, the additional time cost of our method is merely the learning of ReLoss. As described in Section 5.4 of our paper, we analyze the time complexity of ReLoss, and our ReLoss can save much time compared to the previous surrogate loss learning methods [1, 2]. For example, we take 0.5 GPU hours to train the ReLoss on image classification task, while the same loss model yet using alternative optimization as previous methods [1, 2] costs 20 GPU hours (we alternatively train the surrogate loss every 5 epochs on CIFAR-10 in Table 11). Besides, we believe the time cost in ReLoss is negligible compared to the model training time, yet the performance gain is significant. Moreover, it is worth to know that for each task, our ReLoss only need to learn a unified surrogate loss once, then we can adopt it to train all the models, and the additional time cost can thus be further reduced.
>
> **Q2: Will ReLoss increase GPU memory or training time in training?**
>
> **A2**: We believe the memory and training time increments is also negligible compared to the much larger consumptions of models (e.g., ResNet-50 has 25.6M parameters), as our ReLoss only has $\sim0.03$M parameters. In addition, we report the memory consumption and training speed of ResNet-56 on CIFAR-10 and CIFAR-100 datasets as follows.
>
> |Dataset|Loss function|GPU memory (M)|Training speed (batches/second)|ACC (%)|
> |:--:|:--:|:--:|:--:|:--:|
> |CIFAR-10|CE|618.46|14.29|94.32|
> |CIFAR-10|ReLoss|618.68|14.29|**94.57**|
> |CIFAR-100|CE|618.55|14.27|73.61|
> |CIFAR-100|ReLoss|618.76|14.27|**74.15**|
>
> The above table shows that, our ReLoss keep the same training speed as the original CE loss on CIFAR datasets, and the memory consumptions are almost the same (e.g., only 0.22M increase on CIFAR-10).
>
> The consistence of training speed is easy to understand, since our ReLoss takes the same inputs and outputs as traditional loss function, and directly backwards the gradients through the network without manually modifications. Taking some task-specific surrogate loss functions [3,4] as an example of increasing time cost, they usually suffer from high computational cost and inefficient manual gradient assignment. Specifically, AP-loss [3] computes the average precision (AP) for every batch of data and proposes a rule to directly assign the AP value as the gradients of outputs of the model; RaMBO [4] models a list of rank-based metrics (e.g., AP) into a new measurement named rearrangement inequality to optimize, but it also yields an O(nlogn) time complexity of the ranking operation and an additional time cost of assigning manual gradients produced by a combinational solver in [5]. Compared to the above methods, our ReLoss enjoys significant efficiency and can be applied to various tasks.
>
> **References**
>
> [1] Grabocka, Josif, Randolf Scholz, and Lars Schmidt-Thieme. "Learning surrogate losses." arXiv preprint arXiv:1905.10108 (2019).
>
> [2] Patel, Yash, Tomáš Hodaň, and Jiří Matas. "Learning surrogates via deep embedding." In European Conference on Computer Vision, pp. 205-221. Springer, Cham, 2020.
>
> [3] Chen, Kean, Weiyao Lin, Jianguo Li, John See, Ji Wang, and Junni Zou. "AP-loss for accurate one-stage object detection." IEEE Transactions on Pattern Analysis and Machine Intelligence 43, no. 11 (2020): 3782-3798.
>
> [4] Rolínek, Michal, Vít Musil, Anselm Paulus, Marin Vlastelica, Claudio Michaelis, and Georg Martius. "Optimizing rank-based metrics with blackbox differentiation." In Proceedings of the IEEE/CVF Conference on Computer Vision and Pattern Recognition, pp. 7620-7630. 2020.
>
> [5] Pogančić, Marin Vlastelica, Anselm Paulus, Vit Musil, Georg Martius, and Michal Rolinek. "Differentiation of blackbox combinatorial solvers." In International Conference on Learning Representations. 2019.

---

> > ### Comment · Reviewer_eBZN · 2021-11-25
> > **concern about the surrogate loss**
> >
> > Thank you for your clarifications on training efficiency. I still have a concern about the necessity of taking a neural network as the surrogate loss. The proposed ReLoss leverages differentiable ranking algorithm to learn the surrogate loss, which is represented by a neural network. However, [1] shows that the differentiable ranking method can be directly applied to top-k classification tasks by maximizing the rank of positive logits. Compared to [1], why do you choose to train a network with unavoidable optimization errors but not directly adopting the differentiable ranking operator?
> >
> > [1] Mathieu Blondel, Olivier Teboul, Quentin Berthet, and Josip Djolonga. Fast differentiable sorting and ranking. In International Conference on Machine Learning, pp. 950–959. PMLR, 2020.

---

> > > ### Author Response · Authors · 2021-11-29
> > > **Response**
> > >
> > > **Q3: The necessity of taking a neural network as the surrogate loss.**
> > >
> > > **A3:** We thank the reviewer for raising this interesting point. We have implemented the rank-based classification loss in [1], which calculates the L1 error between predicted soft ranks and target ranks of top-1 elements, i.e.,
> > > $$
> > > l_{\text{rk}} = |\textbf{r}_\text{pos}\ –\ N| ,
> > > $$
> > > where $\textbf{r}_\text{pos}$ denotes the predicted soft ranks of $\textbf{y}_\text{pos}$ and N is the number of classes.
> > >
> > > We train the above rank-based classification loss (RankLoss) on CIFAR-10, CIFAR-100, and ImageNet datasets with the same models and strategies in our paper. The validation accuracies are summarized as follows.
> > >
> > > |Dataset|Model|CE Loss (%)|ReLoss (%)|RankLoss (%)|
> > > |:--:|:--:|:--:|:--:|:--:|
> > > |CIFAR-10|ResNet-56|94.32|**94.57**|82.77|
> > > |CIFAR-100|ResNet-56|73.61|**74.15**|5.65|
> > > |ImageNet|ResNet-50|76.5|**76.8**|0.58|
> > >
> > > From the results, we can see that the accuracies obtained by RankLoss are significantly lower than the original CE loss and our ReLoss. Besides, on datasets with more classes (100 and 1000 on CIFAR-100 and ImageNet, respectively), it only obtains slightly better accuracies than random guess. One possible reason is that RankLoss only focuses on the ranks of positive elements of logits, lacking supervision on the remained elements. As a result, the network receives little information to converge on datasets with large numbers of classes. In contrast, our ReLoss learns from the evaluation metrics and supplies better information for discriminating models, thus achieves better accuracy.
> > >
> > > On the other hand, RankLoss is hard to generalize to different tasks since it needs to design different loss functions for different metrics. Meanwhile, it cannot be applied to regression tasks.
> > >
> > > As a result, we believe it is necessary to learn a metric-oriented surrogate loss using a neural network rather than directly applying differentiable ranking operators as the loss.
> > >
> > > **References**
> > >
> > > [1] Mathieu Blondel, Olivier Teboul, Quentin Berthet, and Josip Djolonga. Fast differentiable sorting and ranking. In International Conference on Machine Learning, pp. 950–959. PMLR, 2020.

---

### Official Review · Reviewer_Y6nu · 2021-11-02

**Correctness:** 4
**Technical Novelty And Significance:** 4
**Empirical Novelty And Significance:** 3
**Recommendation:** 8
**Confidence:** 4

**Main Review:**

The proposed approach is well motivated and makes sense. The problem study here is also essential and could be of interest to a large audience, as the loss function is fundamental in almost all machine learning tasks.

Meanwhile, the techniques are simple but effective. Without doubts, the proposed correlation-based optimization that introduces differentiable Spearman's correlation coefficient has looser constraints than approximation-based methods. Also, I believe the gradient penalty is necessary to keep stable convergence and fixed training strategies, since the magnitude of gradients may changes w.r.t. the initialization and randomness.

Experiments are sufficient and convincing. For example, the ReLoss gains noticeable improvements over baselines and outperforms the state-of-the-art method on COCO keypoint dataset.

However, I do have some minor concerns in this paper:

(a) The surrogate losses can be learned with the universal method (ReLoss) without expertise on loss function design, but it seems to introduce a new problem of confirming the architecture of losses w.r.t the evaluation metrics.

(b) Approximation errors always exist as the surrogate loss cannot fully recover the evaluation metrics in the whole distribution. In this paper, the authors adopt the original loss as a regularization term to alleviate this problem, but the superiority of ReLoss on rank correlation may be weakened.

**Summary Of The Paper:**

The authors introduce a relational surrogate loss learning method (ReLoss) for replacing the original losses. The rationale and intuition behind are well-grounded. Experiments on various tasks and ablation studies prove the validity.

**Summary Of The Review:**

The paper proposes a method to learn surrogate losses, compared to related works, the proposed method gains significant improvements on various datasets. The originality and significance are clearly above the bar, though there still remain some minor concerns.

---

> ### Author Response · Authors · 2021-11-22
> **Response to Reviewer Y6nu**
>
> Thank you for the comments and efforts in reviewing our paper. The responses to your comments and questions are as follows.
>
> **Q1: ReLoss may introduce a new problem of designing the architecture of loss model.**
>
> **A1**: Thanks for your comments. Since our ReLoss proposes a less strong constraint (rank correlation) as the learning objective, it is easy to obtain promising correlation and training accuracy using simple neural networks as loss models. Therefore, as in our paper, we construct surrogate losses with mostly the same architectures for tasks with classification heads, for example, only MLP with a few layers will suffice. Meanwhile, as shown in Figure 3 (b) in Appendix, starting from a tiny loss model (~0.03M parameters), our ReLoss can obtain nearly the same accuracies on CIFAR-10, indicating that a tiny and simple model is sufficient to learn the surrogate loss well. Of course, sophisticated design of loss model might further contribute to the performance of proposed ReLoss, but this is not what emphasis in this paper and can be left as future work.
>
> **Q2: The introduction of original loss may weaken the superiority of ReLoss.**
>
> **A2**: Thanks for your concerns. The original loss can be regarded as a regularization term for the ReLoss. Note that original loss is a regular point-to-point supervision to directly pull the loss near the optimal point; in this way, it enables to provide the ReLoss with a good guess or regularization actually, and is expected to further promote the performance. As the results in Table 10, the addition of regular loss can help the ReLoss achieve better performance. For example, on CIFAR-100, training with both ReLoss and regular loss can improve the accuracy with ReLoss only by 0.24%. Note that we did not add the regular loss on text recognition task (Table 7) for fair comparisons, and our ReLoss can still significantly outperform the original loss and the related work LS-ED.

---

### Decision · Program_Chairs · 2022-01-20

**Decision:**

Accept (Poster)

**Comment:**

The paper presents an approach to learn the surrogate loss for complex prediction tasks where the task loss is non-differentiable and non-decomposable. The novelty of the approach is to rely on differentiable sorting, optimizing the spearman correlation between the true loss and the surrogate. This leads to a pipeline that is simpler to integrate to existing works than approaches that try to learn a differentiable approximation to the task loss, and to better experimental results.

The paper is well written and the approach clearly presented. The reviewers liked the simplicity of the approach and the promising experimental results on a variety of challenging tasks (human pose estimation and machine reading).